# Is a Vitamin K Epoxide Reductase Complex Subunit 1 (*VKORC1*) Polymorphism a Risk Factor for Nephrolithiasis in Sarcoidosis?

**DOI:** 10.3390/ijms25084448

**Published:** 2024-04-18

**Authors:** Marjolein Drent, Petal Wijnen, Otto Bekers, Aalt Bast

**Affiliations:** 1ILD Center of Excellence, Department of Respiratory Medicine, St. Antonius Hospital, 3435 CM Nieuwegein, The Netherlands; petal.wijnen@mumc.nl; 2Department of Pharmacology and Toxicology, Faculty of Health, Medicine, and Life Science, Maastricht University, 6200 MD Maastricht, The Netherlands; o.bekers@mumc.nl (O.B.); a.bast@maastrichtuniversity.nl (A.B.); 3Research Team, ILD Care Foundation, 6711 NR Ede, The Netherlands; 4Department of Clinical Chemistry, Central Diagnostic Laboratory, Maastricht University Medical Centre, 6229 HX Maastricht, The Netherlands

**Keywords:** hypercalcemia, kidney stones, nephrolithiasis, sarcoidosis, *VKORC1*, vitamin K

## Abstract

Sarcoidosis is a systemic inflammatory disorder characterized by granuloma formation in various organs. It has been associated with nephrolithiasis. The vitamin K epoxide reductase complex subunit 1 (*VKORC1*) gene, which plays a crucial role in vitamin K metabolism, has been implicated in the activation of proteins associated with calcification, including in the forming of nephrolithiasis. This study aimed to investigate the *VKORC1* C1173T polymorphism (rs9934438) in a Dutch sarcoidosis cohort, comparing individuals with and without a history of nephrolithiasis. Retrospectively, 424 patients with sarcoidosis were divided into three groups: those with a history of nephrolithiasis (Group I: *n* = 23), those with hypercalcemia without nephrolithiasis (Group II: *n* = 38), and those without nephrolithiasis or hypercalcemia (Group III: *n* = 363). Of the 424 sarcoidosis patients studied, 5.4% had a history of nephrolithiasis (Group I), only two of whom possessed no *VKORC1* polymorphisms (OR = 7.73; 95% CI 1.79–33.4; *p* = 0.001). The presence of a *VKORC1* C1173T variant allele was found to be a substantial risk factor for the development of nephrolithiasis in sarcoidosis patients. This study provides novel insights into the genetic basis of nephrolithiasis in sarcoidosis patients, identifying *VKORC1* C1173T as a potential contributor. Further research is warranted to elucidate the precise mechanisms and explore potential therapeutic interventions based on these genetic findings.

## 1. Introduction

Sarcoidosis is an inflammatory, multisystemic condition of uncertain origin that exhibits a variety of clinical manifestations [1]. This disorder can affect nearly any organ in the body but particularly affects the lungs, lymphatic system, skin or eyes, or combinations of these. It is characterized by the development of non-caseating granulomas [1]. Although sarcoidosis is usually not considered a renal condition [2], it can impact the kidneys directly or contribute to renal failure through hypercalcemia, hypercalciuria, and nephrocalcinosis [3]. Screening for hypercalcemia in sarcoidosis has been recommended [3,4]. Nephrolithiasis has been referred to as the first sign for a diagnosis of sarcoidosis in some patients [5].

Hypercalcemia is not specific to sarcoidosis, as it is also observed in other granulomatous and non-granulomatous disorders, including tuberculosis, coccidioidomycosis, histoplasmosis, leprosy, Crohn’s disease, granulomatosis with polyangiitis, hyperpara-thyroidism, and ankylosing spondylitis [6,7]. However, a minority of sarcoidosis patients have a systemic disease causing chronic hypercalcemia and promoting stone formation. In granulomatous disorders, including sarcoidosis, the inappropriate endogenous overproduction of 1,25-dihydroxyvitamin D3 (calcitriol) by activated macrophages and granulomatous tissue is responsible for hypercalcemia and hypercalciuria [6]. Calcitriol, in turn, triggers increased calcium absorption from the gut. Hypercalciuria can be found in up to 50% of patients with sarcoidosis; in comparison, it is found in only 2–5% of healthy adults [6]. Excessive sunlight or vitamin D ingestion may worsen the situation [8,9]. Calcium levels may change according to disease activity or a patient’s total ultra-violet light exposure [5,6]. Hypercalciuria predisposes one to nephrolithiasis and obstructive uropathy. However, nephrolithiasis is a rather rare complication arising from chronic hypercalciuria. It is present in less than 5% of patients living with sarcoidosis [2,5,6].

Nephrolithiasis is regarded as a multifactorial disease associated with systemic diseases such as sarcoidosis [2,10,11]. The pathogenesis of nephrolithiasis or biomineralization is a complex biochemical process that remains incompletely understood [12]. Kidney stone formation is a biological process involving physicochemical changes and the supersaturation of urine [11]. Nephrolithiasis affects individuals of all ages, genders, and ethnicities, although it appears to be more prevalent in men than in women. The majority of kidney stones contain calcium [11,12]. Many factors contribute to calcium oxalate (CaOx) stone formation, including hypercalciuria (resorptive, renal leak, absorptive, and metabolic diseases). Another proposed mechanism is oxidative stress, the common trigger for endothelial dysfunction and chronic inflammation in both nephrolithiasis and vascular disease [11,13,14].

The role of the gut microbiota in nephrolithiasis and the contribution of vitamins, including vitamin K, to urolithiasis are being reconsidered [15]. Vitamers of vitamin K work throughout the body to reduce soft-tissue calcification by activating carboxyglutamate (Gla) proteins, osteocalcin, and Matrix Gla protein (MGP). Extra-hepatic vitamin K status, measured as dephosphorylated uncarboxylated MGP (dpucMGP), maintains vascular health, with high levels reflecting poor vitamin K status [16]. Sufficient concentrations of local vitamin K are required for Gla protein activation; thus, vitamin K2 is most likely responsible for mitigating aberrant calcification [15]. In view of the role of vitamin K2 in local γ-carboxylation, it would be interesting to understand the vitamin K2 statuses of stone formers [15]. Genetic polymorphisms have been proposed to be risk factors affecting the occurrence and recurrence of kidney stones [17]. Previously, it was shown that the expression of the VKORC1 protein is decreased in renal tissues of patients with CaOx urolithiasis, indicating that VKORC1 may play an important role in the formation of CaOx stones [18]. We wondered whether VKORC1 could also play a role in the formation of kidney stones in sarcoidosis, in addition to other effects.

This study aimed to investigate the *VKORC1* C1173T polymorphism in a Dutch cohort of sarcoidosis patients, comparing individuals with and without a history of nephrolithiasis.

## 2. Results

We retrospectively included a total of 424 sarcoidosis patients recruited from two sarcoidosis referral centers in the Netherlands: 154 (36.3%) from the St. Antonius Hospital, Nieuwegein, and 270 (63.7%) from MUMC Maastricht. The patients we studied were subdivided into three groups: those with a history of nephrolithiasis (Group I: *n* = 23; 5.4%), those with hypercalcemia without nephrolithiasis (Group II: *n* = 38; 9.0%), and those without nephrolithiasis or hypercalcemia (Group III: *n* = 363; 85.6%). We did not have access to all the relevant total calcium (TCa) values for Group I. Only 11 TCa values were available at referral, of which 8 were above the normal range of 2.10–2.55 mmol/L. The mean TCa value for Group II was 2.64 mmol/L (2.56–3.20), and that for Group III was 2.36 mmol/L (1.92–2.53). The demographic and clinical data are summarized in Table 1. The prevalence of nephrolithiasis was 5.4%. Of the 269 male patients, 16 had a history of nephrolithiasis (6.3%), while 7 of the 155 women had such a history (4.7%) (*p* = 0.53).

The allele frequencies of the healthy controls derived from the Rotterdam Study (62.1% C) and the frequencies observed in our overall study population as well as in Group III (63.7%/63.6% C, respectively) were similar [19]. However, Group I exhibited reversed frequencies (37.0% C versus 63.0% T), and Group II displayed a higher prevalence of C alleles (80.3%, as indicated in Table 1) [19]. Stratification by gender yielded identical frequencies for C and T alleles in the total sarcoidosis population (C = 40.6% female and 40.5% male, respectively). All Groups were in Hardy–Weinberg equilibrium (see Table 1).

Only two patients from Group I (8.7%) did not possess a *VKORC1* polymorphism (a CC genotype). In contrast, those with only episodes of hypercalcemia (group II) included a significantly higher percentage without polymorphism (64.8%; OR 20.2; 95% CI 4.09–99.8; *p* < 0.0001; Table 1 and Table 2). Of the patients with nephrolithiasis, 4/23 (17.4%) suffered from obesity (Body Mass Index ≥ 30 kg/m^2^). One of the four obese patients also had a history of hypertension. None of the 23 patients used coumarins or vitamin K orally.

Compared with Groups II + III, the patients from Group I with the CT or TT genotype had an almost eight-times-higher probability of developing nephrolithiasis (OR = 7.73; CI 95% 1.79–33.4; *p* = 0.001) than patients without the T-allele. This risk was even higher than the risk for those with hypercalcemia without nephrolithiasis (OR 20.2; *p* < 0.0001; Table 2).

Group I also included significantly fewer normal metabolizers of *VKORC1* than Group II (*p* < 0.0001) and Group III (*p* = 0.003).

Of the patients without a *VKORC1* variant (*n* = 172) in the study sample (*n* = 424), two had a history of nephrolithiasis (1.2%), which can be compared to twenty-one (9.1%) of those with a *VKORC1* variant (252).

Table 3 shows that the CXR stages of those with nephrolithiasis (Group I) differed significantly from those in Group III (OR 2.79; 95% CI 1.19–6.53; *p* = 0.014). Fewer patients with CXR stages II-IV were found, implying less pulmonary involvement. Compared to Group III, Group I (nephrolithiasis) combined with Group II (hypercalcemia) also showed significantly less pulmonary involvement (OR 1.78; 95% CI 1.78–3.11; *p* = 0.042).

## 3. Discussion

In this retrospective study conducted using a Dutch sarcoidosis sample, we found that the prevalence of nephrolithiasis was 5.4%, with a slight predominance in male patients (6.3% versus 4.7%), which, however, was not statistically significant. Additionally, we assessed the association between a *VKORC1* polymorphism implicated in the activation of proteins associated with calcification and nephrolithiasis in patients with sarcoidosis. We found that, compared with Groups II + III (i.e., those without nephrolithiasis), patients from Group I (i.e., those with nephrolithiasis) who possessed the CT or TT genotype had an almost eight-times-higher probability of developing nephrolithiasis (OR 7.73; 95% CI 1.79–33.4; *p* = 0.001) than patients without the T-allele. The presence of a *VKORC1* C1173T variant allele was found to be a substantial risk factor for the development of nephrolithiasis in sarcoidosis patients.

Sarcoidosis is known to be accompanied by changes in calcium metabolism, as well as nephrolithiasis in some cases [20]. Hypercalcemia in sarcoidosis occurs due to the uncontrolled synthesis of 1,25-dihydroxyvitamin D3 by macrophages, which, in turn, leads to increased absorption of calcium in the intestine and increased resorption of calcium in the bone. Nephrolithiasis represents a process of unwanted calcification associated with substantial mortality and high recurrence rates. Numerous factors contribute to this process [11,12]. Reduced functionality of VKORC1 has been correlated with vascular calcification and urological stone formation [21]. The insufficient availability of vitamin K in tissues necessitates prompt reduction of vitamin K epoxide to vitamin K, which subsequently stimulates γ-carboxylation reactions. VKORC1 is the only enzyme responsible for reducing vitamin K epoxide, and it is the rate-limiting factor in vitamin K recycling. It thus acts as the bottleneck enzyme in the γ-carboxylation of vitamin K-dependent proteins. VKORC1 is assumed to play a crucial role in the crystallization of calcium oxalate (CaOx) [22].

The understanding of the pathogenesis of nephrolithiasis, as well as its prevention and cure, still remains rudimentary [11]. The pathogenesis of CaOx stone formation is a multistep process and in essence includes nucleation, crystal growth, crystal aggregation, and crystal retention [2,11,23]. Various substances in the body have an effect on one or more of the above stone-forming processes, thereby influencing the ability to promote or prevent stone formation. Besides low urine volumes and low urine pH, high calcium, sodium, oxalate, and/or urate concentrations are also known to promote CaOx stone formation [11,12]. In addition, many inorganic and organic substances are known to inhibit stone formation. Matrix Gla Protein (MGP), a vitamin-K-dependent extracellular matrix protein, was initially isolated from bone, but it is also expressed in the lungs, heart, vascular smooth muscle cells of the blood vessel walls, and kidneys [16,24]. MGP is a natural inhibitor of vascular calcification [22]. This implies an intricate interplay between vitamin K, coagulation mechanisms, and various facets of health, with the regulation of calcification appearing to play a pivotal role. Understanding these associations is crucial for developing effective interventions and optimizing health outcomes for individuals with various risk factors and conditions. Figure 1 summarizes the influence of sarcoidosis and vitamin K epoxide reductase (VKOR) on the interaction between vitamin K, vitamin D, and calcium. Vitamin K has received considerable attention due to its influence on clotting factors [15]. An expanding body of evidence suggests a correlation between vitamin K status and calcification, impacting bone, heart, and kidney health. The proposed causes of low vitamin K levels include increased nutrient requirements to mitigate calcification, diminished intake associated with an overly restrictive traditional renal diet (particularly potassium restriction), medications such as omeprazole and/or antibiotics that contribute to dysbiosis, and diminishing production of vitamin K by gut bacteria. The use of warfarin is also linked to increased calcification, as well as nephrolithiasis and reduced vitamin K levels, along with uremia, potentially stemming from diminished enzyme activity in the vitamin K cycle [23].

Reduced functionality of VKORC1 is associated with osteoporosis and calcified plaques in the carotid artery [25]. These associations seem to involve different mechanisms. Our patients without a *VKORC1* polymorphism had a lower risk of calcifications. A reduced functionality of VKORC1 may result in vitamin K deficiency [26]. Vitamin K (VK)-dependent g-glutamate carboxylation and serine phosphorylation activate MGP, making it a potent locally acting inhibitor of calcification. Vitamin K2 is capable of activating dpucMGP, which can sequester free calcium and might be able to systemically reduce aberrant calcification [15]. Moreover, it has been shown that vitamin K deficiency is common in kidney stone formers. Sufficient levels of vitamin K help prevent calcium from building up in excess amounts in the blood and along artery walls. Higher levels of inactive dpucMGP may be causally associated with the risk of nephrolithiasis [12]. Research findings suggest that increasing vitamin K levels in one’s diet, or adding vitamin K supplements, may help prevent kidney stones from forming [27]. However, having a less-functional VKORC1 enzyme alone does not necessarily result in vitamin K deficiency. Factors such as diet, the use of certain medications like oral anticoagulants, and exposure to pesticides also contribute to the development of vitamin K deficiency [28].

Understanding the relevant genetic factors may pave the way towards personalized approaches to assessing the risk of nephrolithiasis in individuals with sarcoidosis, and probably also in individuals with other diseases, allowing for targeted preventive strategies and more-effective management of renal complications [17,29]. Renal calculi appear to be a marker of chronicity, with long-term corticosteroid therapy required in most cases [5]. Regardless of the underlying etiology and drug treatment for nephrolithiasis, patients should also be instructed to prevent this disease by drinking more water or other liquids [12]. Sufficient fluid intake reduces urinary saturation and dilutes promoters of CaOx crystallization. Dietary recommendations should be adjusted based on individual metabolic abnormalities [12,23]. A low dietary intake of calcium, vitamin D, and oxalate and avoidance of thiazide-like diuretics (with calcium-sparing proprieties) are generally recommended for patients with hypercalcemia. Limiting sunlight exposure is also advised to prevent increased vitamin D production [9]. In case of hypertension and/or obesity, lifestyle modifications might be beneficial as well [30].

It is known that the use of antibiotics can result in reduced absorption of vitamin K in the intestines, potentially leading to vitamin K deficiency. In a study involving female nurses aged 30 to 55 years, the use of antibiotics for more than two months in early adulthood and at middle age was found to be associated with a higher risk of nephrolithiasis in later life [31]. This is noteworthy because vitamin K deficiency is believed to be a factor in its development. Further study is needed to determine whether vitamin K suppletion can serve as an additional treatment option for those suffering from calcifications such as nephrolithiasis and, more importantly, prevent calcification [23,32].

Recently, it was assumed that sarcoidosis patients with hypercalciuria may have active, reversible interstitial inflammation [33]. Hence, hypercalciuria may predict a better response to immunosuppressive therapy among those with renal sarcoidosis [33]. In the present study, the patients with nephrolithiasis showed less pulmonary involvement (chest X-ray stages 0-I), indicating a favorable prognosis compared with the other sarcoidosis patients. In contrast, another study found that most patients with hypercalcemia had a more chronic course of sarcoidosis [34]. Moreover, an association has been found between nephrolithiasis and pulmonary involvement determined via chest X-ray [5]. A plausible explanation for this difference is that our study cohort comprised patients monitored during the follow-up period and not at the time of initial disease presentation. Consequently, it is conceivable that the reduced pulmonary involvement we observed may, in part, be attributed to the resolution achieved using the treatment administered [33]. An alternative explanation could be rooted in the demographic composition of the population in the other studies, with a higher representation of Black individuals. It is well-established that individuals of Black ethnicity often experience a more severe course of sarcoidosis, characterized by more pronounced pulmonary involvement.

This study has several limitations. The retrospective design of our study may have resulted in incomplete data and data collection bias. However, the prevalence of nephrolithiasis in our sample was comparable to that in other studies [2,5]. Secondly, we did not have access to some relevant clinical data at presentation. No chemical analysis results for kidney stones were available. However, it is well known that the majority of kidney stones are rich in calcium [12]. Furthermore, unfortunately, data associated with vitamin K status (inactive dpucMGP levels) were unavailable. We were not informed about the vitamin K statuses of the sarcoidosis patients. Further studies should involve gather biochemical data, including vitamin D, calcium, and vitamin K levels, prospectively. Moreover, these studies should involve a multivariate analysis that includes these confounding variables.

## 4. Materials and Methods

### 4.1. Patients and Methods

Sarcoidosis patients (*n* = 424) who visited the outpatient referral clinic of the Sarcoidosis Management Center of the Department of Respiratory Medicine of Maastricht University Medical Centre (MUMC), Maastricht, The Netherlands (*n* = 270), or the Interstitial Lung Disease (ILD) Center of Excellence of the Department of Respiratory Medicine of St. Antonius Hospital, Nieuwegein, The Netherlands (*n* = 154), were included in this study. The diagnosis of sarcoidosis was established according to the criteria of the Official American Thoracic Society Clinical Practice Guideline [35]. The characteristics of the sarcoidosis patients, including age at diagnosis, gender, Scadding stage at diagnosis, and information regarding the development of nephrolithiasis, were collected retrospectively from medical records. This study was conducted according to the guidelines of the Declaration of Helsinki and approved by the Medical Ethics Board of the MUMC (METC 11-4-116) and the Medical research Ethics Committees United of the St. Antonius Hospital (MEC-U R05-08A). Written informed consent was obtained from all subjects. 

### 4.2. Collection of Clinical Data

Patients were retrospectively evaluated for having a history of one or more episodes of hypercalcemia (i.e., a total calcium value above the reference value [2.10–2.55 mmol/L]), as well as a history of nephrolithiasis. All chest X-Ray (CXR) films were graded by a single observer, who was not aware of the clinical data. Five stages of radiographical abnormality were identified: (1) (normal CXR), (2) stage I (bilateral hilar lymphadenopathy [BHL]), (3) stage II (BHL and parenchymal abnormalities), (4) stage III (parenchymal abnormalities without BHL), and (5) stage IV (end stage lung fibrosis) [1,35]. For the main analysis, patients were categorized into groups with and without a history of nephrolithiasis.

### 4.3. Genotyping

In all subjects, genomic DNA was isolated from venous EDTA-anticoagulated blood. Genotyping was carried out for the most clinically relevant single-nucleotide polymorphisms (SNPs) of three cytochrome P450 (*CYP*) genes (*CYP2D6*, *CYP2C9*, and *CYP2C19*), as well as *VKORC1*. SNPs were identified via real-time PCR fluorescence resonance energy transfer (FRET) analyses using FRET LightMix® assays (TIB MOLBIOL, Berlin, Germany) via a LightCycler® (Roche Diagnostics, Mannheim, Germany) or via real-time PCR using the StepOnePlus™ Real-Time PCR System and TaqMan GTXpress Master/Drug Metabolizing Genotyping Assay mixes (Applied Biosystems, Foster City, CA, USA), according to the manufacturer’s instructions.

In accordance with conventional classification systems, individuals were classified as poor metabolizers (PMs) if they carried two non-functional alleles; intermediate metabolizers (IMs) if they carried one non-functional allele; or normal metabolizers (NMs) if they carried two functional alleles.

### 4.4. Statistical Analysis and Risk Stratification

Statistical analyses were performed with SPSS version 28.0 software for Windows (SPSS Inc., Chicago, IL, USA). Chi-square statistical analysis was used to test for differences between groups. Odds ratios (ORs) with 95% confidence intervals (CIs) were calculated to evaluate the strength of associations between genotypes and the development of nephrolithiasis. Actual allele distributions were compared with the expected frequencies calculated using the Hardy–Weinberg equilibrium. A *p*-value < 0.05 was considered statistically significant. Where necessary, a Bonferroni correction was applied (*p*-value < 0.01). To compare the distribution of polymorphisms between the patients studied and the general Caucasian population, controls were collected from the literature [19].

## 5. Conclusions

The prevalence of nephrolithiasis in a Dutch sample of patients with sarcoidosis was 5.4%. We found a significant association between the *VKORC1* C1173T polymorphism and nephrolithiasis in sarcoidosis patients. The presence of a *VKORC1* variant allele may serve as a potential genetic marker of the risk of developing nephrolithiasis in conjunction with pre-existing sarcoidosis. This study provides novel insights into the genetic basis of nephrolithiasis among sarcoidosis patients, implicating *VKORC1* C1173T as a potential contributor. Further research is warranted to elucidate the precise mechanisms, including the role of vitamin K deficiency, and to explore potential therapeutic interventions based on these genetic findings, e.g., vitamin K suppletion and diet and/or medication changes. Also, it would be interesting to understand the *VKORC1* polymorphism in patients with nephrolithiasis without sarcoidosis.

## Figures and Tables

**Figure 1 ijms-25-04448-f001:**
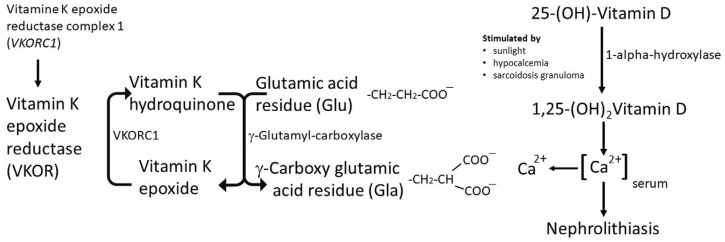
The influence of sarcoidosis and vitamin K epoxide reductase (VKOR) on the interaction between vitamin K, vitamin D, and calcium. The active form of vitamin K is vitamin K hydroquinone. This is formed via vitamin K epoxide reductase complex subunit 1 (VKORC1). Vitamin K hydroquinone activates the enzyme g-glutamyl carboxylase, which leads to the carboxylation of glutamic acid residues, enabling proteins to form a complex with calcium ions (Ca^2+^). Hydroxylation of 25-(OH)-vitamin D produces 1,25-(OH)_2_—vitamin D, resulting in calcium mobilization. The presence of calcium ions leads to nephrolithiasis.

**Table 1 ijms-25-04448-t001:** Summary of demographic and clinical data of the sarcoidosis groups studied [19].

	Number	Gender m/f	Age y; Mean	CXR-Stage 0–I/II–IV	*VKORC1* CC/CT/TT	*VKORC1* No var/var	*VKORC1* HW *p*-Value; freq C/T
Group I	23 (5.4)	16/7 (70.8/29.2)	41 (27–60)	12/11 (52.2/47.8)	2/13/8	2/21 (8.7/91.3)	0.31 (37.0/63.0)
Group II	38 (9.0)	26/12 (68.4/31.6)	41 (21–65)	13/25 (34.2/65.8)	25/11/2	25/13 (65.8/24.2)	0.59 (80.3/19.7)
Group III	363 (85.6)	227/136 (62.5/37.5)	43 (13–82)	102/261 (28.1/71.9)	145/172/46	145/218 (39.9/60.1)	0.65 (63.6/36.4)
Total	424 (100.0)	269/155 (63.4/36.6)	43 (13–82)	127/297 (30.0/70.0)	172/196/56	172/252 (40.6/59.4)	0.99 (63.7/36.3)

Data are expressed as absolute numbers, with percentages or ranges, if appropriate, given in parentheses. Abbreviations: f = female; m = male; y = year; CXR = X-ray stage; *VKORC1* = vitamin K epoxide reductase complex subunit 1 C1173T; var = variant allele (CT or TT); HW = Hardy–Weinberg; freq = allele frequency. Group I = nephrolithiasis; Group II = hypercalcemia without nephrolithiasis; Group III = sarcoidosis without hypercalcemia or nephrolithiasis; healthy controls from the Rotterdam study (C = 62.1% and T = 37.9%) [19].

**Table 2 ijms-25-04448-t002:** Probability of developing nephrolithiasis associated with possessing vitamin K epoxide reductase complex subunit (*VKORC1*) C1173T for the sarcoidosis groups studied.

	*VKORC1* No var vs. var	OR	95% CI	*p*-Value
Group I vs. II	2/21 vs. 25/13	20.2	4.09–99.8	**<0.0001**
Group I vs. III	2/21 vs. 145/218	6.98	1.61–30.2	**0.003**
Group I vs. II + III	2/21 vs. 170/231	7.73	1.79–33.4	**0.001**
Group I + II vs. III	27/34 vs. 145/218	0.84	0.48–1.45	0.527
Group II vs. III	25/13 vs. 145/218	0.35	0.17–0.70	**0.002**

Abbreviations: *p* = *p*-value (*p* < 0.05 is significant), where bold indicates significance; CI = confidence interval; OR = odds ratio; vs. = versus; *VKORC1* = vitamin K epoxide reductase complex subunit 1 C1173T; var = variant allele (CT or TT); Group I = nephrolithiasis; Group II = hypercalcemia without nephrolithiasis; Group III = sarcoidosis without hypercalcemia or nephrolithiasis.

**Table 3 ijms-25-04448-t003:** Probability of no pulmonary involvement (chest X-ray stages 0–I) for the sarcoidosis groups studied.

	CXR 0 + I vs. II–IV	OR	95% CI	*p*-Value
Group I vs. II	12/11 vs. 13/25	2.10	0.73–6.04	0.17
Group I vs. III	12/11 vs. 102/261	2.79	1.19–6.53	**0.014**
Group I vs. II + III	12/11 vs. 115/286	2.71	1.16–6.32	**0.017**
Group I + II vs. III	25/36 vs. 102/261	1.78	1.78–3.11	**0.042**
Group II vs. III	13/25 vs. 102/261	1.33	0.66–2.70	0.43

Abbreviations: *p* = *p*-value (*p* < 0.05 is significant), where bold indicates significance; CI = confidence interval; OR = odds ratio; vs. = versus; CXR = X-ray stage; Group I = nephrolithiasis; Group II = hypercalcemia without nephrolithiasis; Group III = sarcoidosis without hypercalcemia or nephrolithiasis.

## Data Availability

The data presented in this study are available upon request from the corresponding author.

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
