# Peer review of "Is a Vitamin K Epoxide Reductase Complex Subunit 1 (VKORC1) Polymorphism a Risk Factor for Nephrolithiasis in Sarcoidosis?"

_ijms, 2024, doi:10.3390/ijms25084448_

Round 1
Reviewer 1 Report
Comments and Suggestions for Authors
The manuscript "Is a Vitamin K Epoxide Reductase Complex Subunit 1 (VKORC1) Polymorphism a Risk Factor for Nephrolithiasis in Sarcoidosis?" is a well designed and very interesting study to emphasise the role of vitamin K epoxide reductase complex subunit 1 (VKORC1) gene in vitamin K metabolism for evaluating the Nephrolithiasis in Sarcoidosis. All section are scientifically sound including the patient number and grouping. The method of study was also described adequately. Discussion section also well cited and could be interesting for the readers of medical sciences.
Author Response
Response to reviewer 1.
COMMENTS FOR THE AUTHOR:
The manuscript "Is a Vitamin K Epoxide Reductase Complex Subunit 1 (VKORC1) Polymorphism a Risk Factor for Nephrolithiasis in Sarcoidosis?" is a well-designed and very interesting study to emphasise the role of vitamin K epoxide reductase complex subunit 1 (VKORC1) gene in vitamin K metabolism for evaluating the Nephrolithiasis in Sarcoidosis. All section are scientifically sound including the patient number and grouping. The method of study was also described adequately. Discussion section also well cited and could be interesting for the readers of medical sciences.
We would like to thank the reviewer for this positive response and appreciation of our manuscript
Reviewer 2 Report
Comments and Suggestions for Authors
The researchers retrospectively examined a cohort of 424 individuals afflicted with sarcoidosis, reporting an association between the VKORC1 C1173T polymorphism and the occurrence of nephrolithiasis. The premise behind the investigation is logically sound, and the statistical methods employed are deemed acceptable. Nonetheless, several issues detract from the study's overall import, prompting me to offer several comments to enhance the manuscript's academic merit.
Major
#1 The patient's background needs to be more detailed. Given the robust association between obesity and nephrolithiasis, it is advisable to document the Body Mass Index (BMI) and history of hypertension. Furthermore, it is recommended to include data pertaining to vitamin K and warfarin po, should such information be accessible. In addition, the authors’ prior admission of this in the limitations section, enriching the manuscript with biochemical indices about vitamin D, calcium, and vitamin K levels, might substantially ameliorate this work.
#2 In Table 1's conventional structuring, patient backgrounds are stratified by either the presence or degree of exposure. The authors separate them by the presence of nephrolithiasis, which is the outcome. However, would it not be better to list them separately based on the presence of genetic mutations, which is the exposure?
#3 In a non-randomized clinical investigation like the present study, it is advisable to undertake a multivariate analysis that includes confounding variables to dissect the association between exposure and outcome. Conducting such an analysis might have been impractical due to the number of outcome cases, and while the analytical methods utilized are considered suitable, mentioning this point in the manuscript could enhance its transparency.
Minor
#1 Were cases with a history of hypercalcemia included in Group 1 with a history of nephrolithiasis?
#2 In the abstract, it is difficult to understand whether patients in group 2 have a history of nephrolithiasis.
Comments on the Quality of English LanguageThe researchers retrospectively examined a cohort of 424 individuals afflicted with sarcoidosis, reporting an association between the VKORC1 C1173T polymorphism and the occurrence of nephrolithiasis. The premise behind the investigation is logically sound, and the statistical methods employed are deemed acceptable. Nonetheless, several issues detract from the study's overall import, prompting me to offer several comments to enhance the manuscript's academic merit.
Major
#1 The patient's background needs to be more detailed. Given the robust association between obesity and nephrolithiasis, it is advisable to document the Body Mass Index (BMI) and history of hypertension. Furthermore, it is recommended to include data pertaining to vitamin K and warfarin po, should such information be accessible. In addition, the authors’ prior admission of this in the limitations section, enriching the manuscript with biochemical indices about vitamin D, calcium, and vitamin K levels, might substantially ameliorate this work.
#2 In Table 1's conventional structuring, patient backgrounds are stratified by either the presence or degree of exposure. The authors separate them by the presence of nephrolithiasis, which is the outcome. However, would it not be better to list them separately based on the presence of genetic mutations, which is the exposure?
#3 In a non-randomized clinical investigation like the present study, it is advisable to undertake a multivariate analysis that includes confounding variables to dissect the association between exposure and outcome. Conducting such an analysis might have been impractical due to the number of outcome cases, and while the analytical methods utilized are considered suitable, mentioning this point in the manuscript could enhance its transparency.
Minor
#1 Were cases with a history of hypercalcemia included in Group 1 with a history of nephrolithiasis?
#2 In the abstract, it is difficult to understand whether patients in group 2 have a history of nephrolithiasis.
Author Response
Response to reviewer 2.
We thank you for your constructive and valuable comments. We address each of your points below.
GENERAL COMMENTS FOR THE AUTHOR:
The researchers retrospectively examined a cohort of 424 individuals afflicted with sarcoidosis, reporting an association between the VKORC1 C1173T polymorphism and the occurrence of nephrolithiasis. The premise behind the investigation is logically sound, and the statistical methods employed are deemed acceptable. Nonetheless, several issues detract from the study's overall import, prompting me to offer several comments to enhance the manuscript's academic merit.
Major comments 1:
The patient’s background needs to be more detailed. Given the robust association between obesity and nephrolithiasis, it is advisable to document the Body Mass Index (BMI) and history of hypertension.
In accordance with your suggestion, we added the following to the Result section:
‘Of the patients with nephrolithiasis, 4/23 (17.4%) suffered from obesity (Body Mass Index ≥30kg/m2). One of the four obese patients also had a history of hypertension.’
We also added the following to the Discussion section: ‘In case of hypertension and/or obesity, lifestyle modifications might be beneficial as well [1].’
Furthermore, it is recommended to include data pertaining to vitamin K and warfarin po, should such information be accessible.
None of the 23 patients used coumarins or vitamin K po. We added the following to the Result section: ‘None of the 23 patients used coumarins or vitamin K orally.’
In addition, the authors’ prior admission of this in the limitations section, enriching the manuscript with biochemical indices about vitamin D, calcium, and vitamin K levels, might substantially ameliorate this work.
We agree with the reviewer that detailed information about biochemical indices about vitamin D, calcium, and vitamin K levels would strengthen the academic merit. However, as this was a retrospective study, and more importantly most patients were referred from other hospitals, we do not have access to these data. To acknowledge and address this important comment we added the following to the Discussion section: ‘Further studies should include gathering biochemical data, including vitamin D, calcium, and vitamin K levels. Moreover, these studies should involve a multivariate analysis that includes these confounding variables.’
Comment 2:
In Table 1's conventional structuring, patient backgrounds are stratified by either the presence or degree of exposure. The authors separate them by the presence of nephrolithiasis, which is the outcome. However, would it not be better to list them separately based on the presence of genetic mutations, which is the exposure?
The aim of this study was to evaluate differences in the VKORC1 C1173T polymorphism between Dutch sarcoidosis patients with nephrolithiasis and those without. If we were to rank the sarcoidosis patients we studied according to the presence of VKORC1 variant alleles, the 23 nephrolithiasis patients would stand out less from the total population of 424 cases. This is why we decided to present the results in this way. To highlight the genetic mutations underlying the phenotype of sarcoidosis patients with nephrolithiasis we prefer to keep this structure.
Comment 3:
In a non-randomized clinical investigation like the present study, it is advisable to undertake a multivariate analysis that includes confounding variables to dissect the association between exposure and outcome. Conducting such an analysis might have been impractical due to the number of outcome cases, and while the analytical methods utilized are considered suitable, mentioning this point in the manuscript could enhance its transparency.
We agree that undertaking a multivariate analysis that includes confounding variables to dissect the association between exposure and outcome would enhance the manuscript's academic merit. However, these data are not available due to the retrospective design. A multivariate analysis that includes these confounding variables should be conducted in future studies.
We added the following to the limitations section:
‘Further studies should include gathering biochemical data, including vitamin D, calcium, and vitamin K levels. Moreover, these studies should involve a multivariate analysis that includes these confounding variables.’
Minor Comments 1:
Were cases with a history of hypercalcemia included in Group 1 with a history of nephrolithiasis?
Yes. At least nearly half of the patients had a known history of hypercalcemia.
Minor Comments 2:
In the abstract, it is difficult to understand whether patients in group 2 have a history of nephrolithiasis.
We clarified this inconvenience by adding: …those with hypercalcemia without nephrolithiasis (Group II: n=38),…
Reviewer 3 Report
Comments and Suggestions for Authors
The research conducted by Marjolein Drent et al. presents a thorough investigation into the potential relationship between VKORC1 polymorphism and nephrolithiasis among sarcoidosis patients. The study's focus is relevant, and the manuscript adheres to academic standards regarding structure, language usage, and citation. However, a notable concern raised within the paper is the absence of patients' clinical data such as vitamin K status, serum calcium, parathyroid hormone, phosphorus levels, and kidney function, although the authors acknowledge these limitations. Despite this significant drawback, I find the manuscript to be innovative and important, deserving of publication in the form of a communication-type of the manuscript. The only recommendation I propose is to include a study flow chart at the beginning of the Results section, providing a clear overview of the eligible and included patients in each group. Additionally, please verify the percentage of patients in each group mentioned in lines 89-90.
Author Response
Response to reviewer 3.
We thank you for valuable comments. We address each of your points below.
GENERAL COMMENTS FOR THE AUTHOR:
The research conducted by Marjolein Drent et al. presents a thorough investigation into the potential relationship between VKORC1 polymorphism and nephrolithiasis among sarcoidosis patients. The study's focus is relevant, and the manuscript adheres to academic standards regarding structure, language usage, and citation. However, a notable concern raised within the paper is the absence of patients' clinical data such as vitamin K status, serum calcium, parathyroid hormone, phosphorus levels, and kidney function, although the authors acknowledge these limitations. Despite this significant drawback, I find the manuscript to be innovative and important, deserving of publication in the form of a communication-type of the manuscript.
We would like to thank the reviewer for this positive response and appreciation of our manuscript.
Comment 1:
The only recommendation I propose is to include a study flow chart at the beginning of the Results section, providing a clear overview of the eligible and included patients in each group.
Reading the beginning of the Result section again we think it was confusing first to mention the following: ‘The prevalence of nephrolithiasis was 5.4%. Of the 269 male patients, 16 had a history of nephrolithiasis (6.3%), compared with 7 of the 155 women (4.7%); p=0.53.
Therefore, we changed the beginning with providing an overview of the eligible and included patients in each group: ‘The patients we studied were subdivided into three groups: those with a history of nephrolithiasis (Group I: n=23; 5.4%), those with hypercalcemia without nephrolithiasis (Group II: n=38; 9.0%) and those without nephrolithiasis or hypercalcemia (Group III: n=363; 85.6%). We hope the subdivision into the studied groups is clearer now and believe that adding a flowchart wouldn't provide significant additional value.
Comment 2:
Additionally, please verify the percentage of patients in each group mentioned in lines 89-90.’
According to your suggestion we added the percentage of patients in each group mentioned in lines 89-90. We retrospectively included a total of 424 sarcoidosis patients, recruited from two sarcoidosis referral centers in the Netherlands: 154 (36.3%) from the St. Antonius Hospital, Nieuwegein and 270 (63.7%) from MUMC Maastricht.
Reviewer 4 Report
Comments and Suggestions for Authors
This study is intriguing as it explores the genetic foundations of nephrolithiasis (kidney stones) in patients with sarcoidosis, a systemic inflammatory disorder. It examines the role of a specific genetic variant, VKORC1 C1173T, in increasing the risk of stone formation. The study's findings enhance our understanding of how different diseases may interact and influence each other. Furthermore, by illuminating the genetic aspect of nephrolithiasis in sarcoidosis patients, the study introduces possibilities for new therapeutic interventions based on genetic findings. Such interventions could potentially prevent or treat kidney stones in this patient population, improving their quality of life.
1. It's worth noting that despite the intriguing problem formulation and significant data obtained, the work lacks an analysis of certain aspects of nephrolithiasis and its correlation with the considered polymorphism. Specifically, the authors don't mention their own or others' studies of similar polymorphism in patients without identified sarcoidosis. The study only examined a cohort of patients with sarcoidosis, differing solely in the presence of urolithiasis and/or hypercalcemia (or without these manifestations). It would be interesting to understand the VKORC1 polymorphism in healthy individuals, as well as in patients with nephrolithiasis without sarcoidosis. If the authors deem it unnecessary to consider such cohorts, this should be clearly explained.
2. The work clearly demonstrates a certain sex-specificity of diseases studied. It's necessary to provide data on the distribution of polymorphism (or no variation) depending on the patients' gender. Adding this data to the tables would be beneficial.
3. Data on hypercalcemia is not provided. Although it's mentioned that some patients had elevated calcium, direct data on this parameter's value is not provided. This should be done for all patient groups.
4. I wonder if it's possible to find any dependencies on taking vitamin supplements containing vitamin K, or on changes in diet/medication in the disease's manifestation?
Comments on the Quality of English LanguageEnglish needs some improvement.
Author Response
Response to reviewer 4.
We thank you for your constructive and valuable comments. We address each of your points below.
GENERAL COMMENTS FOR THE AUTHOR:
This study is intriguing as it explores the genetic foundations of nephrolithiasis (kidney stones) in patients with sarcoidosis, a systemic inflammatory disorder. It examines the role of a specific genetic variant, VKORC1 C1173T, in increasing the risk of stone formation. The study's findings enhance our understanding of how different diseases may interact and influence each other. Furthermore, by illuminating the genetic aspect of nephrolithiasis in sarcoidosis patients, the study introduces possibilities for new therapeutic interventions based on genetic findings. Such interventions could potentially prevent or treat kidney stones in this patient population, improving their quality of life.
Comment 1:
It's worth noting that despite the intriguing problem formulation and significant data obtained, the work lacks an analysis of certain aspects of nephrolithiasis and its correlation with the considered polymorphism. Specifically, the authors don't mention their own or others' studies of similar polymorphism in patients without identified sarcoidosis. The study only examined a cohort of patients with sarcoidosis, differing solely in the presence of urolithiasis and/or hypercalcemia (or without these manifestations). It would be interesting to understand the VKORC1 polymorphism in healthy individuals, as well as in patients with nephrolithiasis without sarcoidosis. If the authors deem it unnecessary to consider such cohorts, this should be clearly explained.
We used the Rotterdam Study to illustrate VKORC1 allele frequencies in healthy controls. This study investigated calcification and not sarcoidosis, and tested the VKORC1 C1173T SNP. We have now clarified this in the Result section as follows: ‘The allele frequencies of healthy controls derived from the Rotterdam Study (62.1% C) and the frequencies observed in our overall study population as well as in Group III (63.7% /63.6% C, respectively) were similar [19].’
We agree with the reviewer that it would be very interesting to perform a study in a population of patients with nephrolithiasis without sarcoidosis. So far, however, no such data are available.
We have acknowledged this by adding the following to the Conclusion section: ‘Also, it would be interesting to understand the VKORC1 polymorphism in patients with nephrolithiasis without sarcoidosis.’
Comment 2:
The work clearly demonstrates a certain sex-specificity of diseases studied. It's necessary to provide data on the distribution of polymorphism (or no variation) depending on the patients' gender. Adding this data to the tables would be beneficial.
The study population contained more male than female sarcoidosis patients.
However, no difference between male and female patients were found regarding the VKORC1 C1173T SNP.
|
424 |
CC |
CT |
TT |
|
|
155 female |
63 |
73 |
19 |
|
|
% |
40.6 |
47.1 |
12.3 |
59.4 |
|
269 male |
109 |
123 |
37 |
|
|
% |
40.5 |
45.7 |
13.8 |
59.5 |
To clarify this we added the following to the Result section: ‘Stratification by gender yielded identical allele frequencies for C and T alleles (C =40.6% female and 40.5% male, respectively).’
Comment 3:
Data on hypercalcemia is not provided. Although it's mentioned that some patients had elevated calcium, direct data on this parameter's value is not provided. This should be done for all patient groups.
This study had a retrospective design. This explains why we did not have access to these data, unfortunately. Therefore, we added the following to the Discussion section: ‘Future studies should involve gathering biochemical data, including vitamin D, calcium, and vitamin K levels.’
Comment 4:
I wonder if it's possible to find any dependencies on taking vitamin supplements containing vitamin K, or on changes in diet/medication in the disease's manifestation?
Mainly due to the retrospective design of our study, the relevant data are lacking, as was also mentioned in the Discussion section. Although information about supplements was thus lacking, supplements taken by patients don’t usually include vitamin K. It might be possible that the patients had made changes to their diet. However, it is highly likely that these were made after the presentation of the nephrolithiasis. It would be very interesting to study whether diet changes and/or vitamin K supplementation can influence the occurrence of nephrolithiasis.
Comments on the Quality of English Language
English needs some improvement.
We have asked a professional English Language editor to check the language.
[1] Obligado SH, Goldfarb DS. The association of nephrolithiasis with hypertension and obesity: a review. Am J Hypertens. 2008 Mar;21(3):257-64.
Round 2
Reviewer 4 Report
Comments and Suggestions for Authors To most of my comments, the authors replied that it was impossible to perform additional experiments or additional data analysis. According to the authors, this is because their study is retrospective and they do not have access to data on these patients other than those already mentioned in the paper. I believe that this significantly reduces the value of this study, as the questions I asked were fundamental to the interpretation of the available data. In any case, statements in the methods, discussions and abstraction should clearly indicate the limitations of the study related to its retrospectivity and the impossibility of obtaining additional data. Comments on the Quality of English LanguageI have no comments
Author Response
Reply reviewer 4 to comments of the authors:
To most of my comments, the authors replied that it was impossible to perform additional experiments or additional data analysis. According to the authors, this is because their study is retrospective and they do not have access to data on these patients other than those already mentioned in the paper.
I believe that this significantly reduces the value of this study, as the questions I asked were fundamental to the interpretation of the available data.
In any case, statements in the methods, discussions and abstraction should
clearly indicate the limitations of the study related to its retrospectivity and the impossibility of obtaining additional data.
Comments from the authors:
We apologize that our response did not meet your expectations. We understand the importance of calcium values. Upon reviewing your questions and comments, we believe that it is particularly our response to your third comment which you found inadequate.
You requested us to provide the calcium values for all groups. Since we cannot do so comprehensively, we initially refrained from doing so at all. However, as you still emphasize their importance, we have added some information to the Results section. We did not have the calcium values for the majority of Group I at the time when the patients had nephrolithiasis. However, we know that 8/11 of these patients had hypercalcemia at least once. Unfortunately, the data for the remaining 12 patients cannot be determined, as kidney stone formation occurred before referral to one of the two expert centres. We have added the following to the Results section: ‘We did not have access to all relevant total calcium (TCa) values for Group I. Only 11 TCa values were available at referral, of whom eight were above the normal range of 2.10-2.55 mmol/L. The mean TCa for Group II was 2.64 mmol/L (2.56-3.20) and that for Group III 2.36 mmol/L (1.92-2.53)’
Perhaps it is also worth mentioning that we have already added the following sentence to the Limitations section, in response to a comment from reviewer 2: ‘Further studies should include gathering biochemical data, including vitamin D, calcium, and vitamin K levels prospectively. Moreover, these studies should involve a multivariate analysis that includes these confounding variables.
Comment 4:
I wonder if it's possible to find any dependencies on taking vitamin supplements containing vitamin K, or on changes in diet/medication in the disease's manifestation? Mainly due to the retrospective design of our study, the relevant data are lacking, as was also mentioned in the Discussion section. Although information about supplements was thus lacking, supplements taken by patients don’t usually include vitamin K. It might be possible that the patients had made changes to their diet. However, it is highly likely that these were made after the presentation of the nephrolithiasis. It would be very interesting to study whether diet changes and/or vitamin K supplementation can influence the occurrence of nephrolithiasis.
To clarify this, we added the following to the Conclusion section: Further research is warranted to elucidate the precise mechanisms, including the role of vitamin K deficiency, and to explore potential therapeutic interventions based on these genetic findings, e.g. vitamin K suppletion and diet and/or medication changes.
Round 3
Reviewer 4 Report
Comments and Suggestions for Authors
I have no additional comments